# Multiplexed 3D super-resolution imaging of whole cells using spinning disk confocal microscopy and DNA-PAINT

Florian Schueder[1,2], Juanita Lara-Gutiérrez[3,4], Brian J. Beliveau[3,4], Sinem K. Saka[3,4], Hiroshi M. Sasaki[3,4], Johannes B. Woehrstein[1,2], Maximilian T. Strauss[1,2], Heinrich Grabmayr[1,2], Peng Yin[3,4] & Ralf Jungmann[1,2]

Single-molecule localization microscopy (SMLM) can visualize biological targets on the nanoscale, but complex hardware is required to perform SMLM in thick samples. Here, we combine 3D DNA points accumulation for imaging in nanoscale topography (DNA-PAINT) with spinning disk confocal (SDC) hardware to overcome this limitation. We assay our achievable resolution with two- and three-dimensional DNA origami structures and demonstrate the general applicability by imaging a large variety of cellular targets including proteins, DNA and RNA deep in cells. We achieve multiplexed 3D super-resolution imaging at sample depths up to ~10 μm with up to 20 nm planar and 80 nm axial resolution, now enabling DNA-based super-resolution microscopy in whole cells using standard instrumentation.

---

[1] Department of Physics and Center for Nanoscience, Ludwig Maximilian University, 80539 Munich, Germany. [2] Max Planck Institute of Biochemistry, 82152 Martinsried, Germany. [3] Wyss Institute for Biologically Inspired Engineering, Boston, MA 02115, USA. [4] Department of Systems Biology, Harvard University, Boston, MA 02115, USA. Correspondence and requests for materials should be addressed to P.Y. (email: py@hms.harvard.edu) or to R.J. (email: jungmann@biochem.mpg.de)

The recent development of super-resolution imaging methods has transformed the role played by fluorescence microscopes in biology[1]. These techniques allow researchers to overcome the classical diffraction limit of light while maintaining many of the key benefits of fluorescence microscopy, such as compatibility with a broad range of sample types and the ability to readily multiplex. Popular super-resolution methods include stimulated emission depletion (STED) microscopy[2], structured illumination microscopy (SIM)[3], and single-molecule localization microscopy (SMLM) techniques such as stochastic optical reconstruction microscopy (STORM)[4, 5] and photoactivated localization microscopy (PALM)[6]. SMLM approaches in particular can offer resolutions of ~5–10 nm planar[7] and ~20 nm axial[8]. While the stochastic switching in STORM and PALM is enabled by modulating the emission (e.g., to achieve distinct molecular states, similar to different "colors"[9]) of fixed, target-bound organic fluorophores or fluorescent proteins, a different approach called points accumulation for imaging in nanoscale topography (PAINT)[10] (and related methods[11, 12]) rely on freely diffusing dyes or dye-labeled entities that either bind statically (and emit from there until they bleach) to target molecules or transiently interact with them. SMLM techniques rely on the observation of single-molecule fluorescence events and require a high signal-to-noise ratio (SNR) in order to enable detection of the fluorescence emission of individual molecules. Consequently, the implementation of SMLM has largely been restricted to selective illumination configurations such as total internal reflection (TIRF[13], only offering very limited sample

penetration depths of a few hundred nanometers) or highly inclined and laminated optical sheets (HILO[14]) (Fig. 1a). Furthermore, most PAINT-type SMLM approaches, although relatively straightforward to implement, use non-fluorogenic probes, limiting efficient single-molecule detection without selective illumination due to elevated background fluorescence.

Confocal microscopy, which uses pinholes to physically block out-of-focus light, offers an alternative strategy for performing high SNR imaging with deep sample penetration. In particular, spinning disk confocal (SDC) technology, which employs a parallel array of pinholes on a rotating disk (Fig. 1b), provides an alternative to point-scanning confocal imaging, offering confocal sectioning capabilities while maintaining the advantage of using cameras as spatial detectors (similar to a wide-field system). Recent studies have demonstrated the general ability to perform the SMLM techniques STORM or PALM with an SDC setup[15, 16]. However, the achievable spatial resolution and imaging quality is limited by the very mechanism of achieving photoswitching in these methods, as the disk in an SDC microscope reduces both the excitation intensity as well as the detection efficiency. In STORM, this impacts the ability to switch dyes into the dark state (as this is a photoinduced effect), limiting the applicability to areas with low labeling densities, as most fluorescent molecules need to reside in the dark state in any given imaging frame to allow for efficient single-molecule detection at such targets. While this is less of a concern for PALM microscopy, bleaching of fluorescent proteins outside of the detection volume prior to detection in the confocal volume

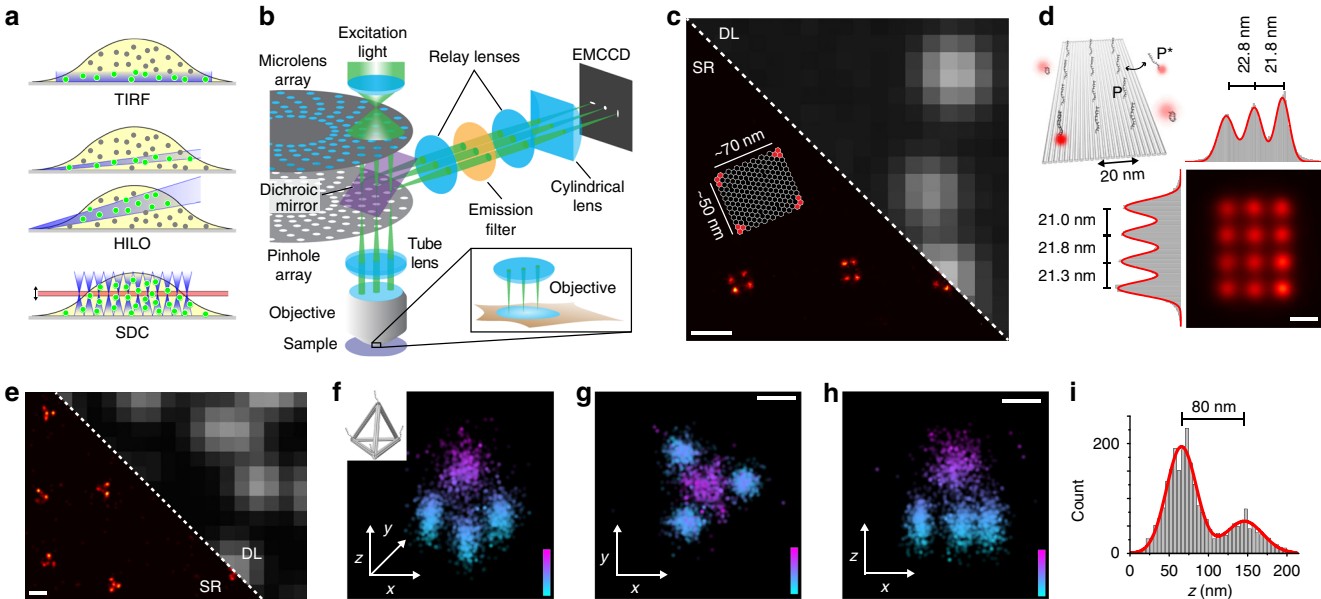

**Fig. 1** DNA-PAINT with sub-15-nm lateral and sub-50-nm axial resolution using spinning disk confocal (SDC) microscopy. **a** Different illumination and detection schemes. Total internal reflection fluorescence (TIRF) creates an evanescent field at the surface. Highly inclined and laminated optical sheet (HILO) enables sheet-like illumination for deeper imaging. Spinning disk confocal (SDC) detects emission light only from the focal plane (red). **b** Schematic drawing of an SDC where 3D super-localization is achieved by optical astigmatism through a cylindrical lens in the detection path. **c** DNA origami structures imaged with DNA-PAINT on an SDC. Diffraction-limited (DL) and super-resolution (SR) representation. The schematic inset shows the DNA origami structure with docking strands in each corner. **d** SDC–DNA-PAINT allows sub-20 nm resolution. Schematic depiction of the DNA origami structure (top left). Super-resolution sum image of 1028 structures (bottom right) along with histograms of columns and rows of the DNA origami. The measured distances are in good agreement with the designed distances. **e** Lateral 2D projection of DNA origami tetrahedrons (designed length of each edge is 100 nm). Diffraction-limited image (DL) and DNA-PAINT super-resolved image (SR). **f** Isometric 3D sum image of 42 tetrahedrons (inset: design of tetrahedron). This panel represents an isometric 3D sum image as overview image of the 3D tetrahedron. **g** x–y projection of the summed image. **h** x–z projection of the summed image. **i** Height measurement of the tetrahedrons obtained from the cross-sectional histogram in the x–z projection from **h** yields 80 nm. Scale bars, 200 nm (**c**, **e**), 20 nm, **d**, 50 nm (**g**, **h**). Height scale, 0–200 nm (**f–h**)

will here ultimately reduce the number of detectable proteins. Additionally, because the disk also blocks part of the emission light, the number of detected photons is reduced, in turn resulting in rather low localization precisions and thus limited achievable spatial resolution for both techniques.

Here, we introduce a platform for SMLM that combines SDC microscopy with DNA-PAINT[17, 18], a recently developed implementation of the PAINT concept in which the transient binding of dye-labeled DNA strands (henceforth called "imager" strands) to complementary anchor strands on the targets to be imaged produces observable single-molecule fluorescence events. Importantly, as the single-molecule fluorescence events are driven by DNA hybridization, the on- and off-rates can be precisely tuned, thus allowing the detectable photons from a single "on" (or binding event) to be maximized and the "switching" rate to be dynamically matched to the density of target sites in the sample. We further demonstrate the multiplexed imaging capacity of this approach for a broad range of cellular targets, and introduce the ability to achieve lateral and axial super-resolution using astigmatism-based 3D SMLM in combination with an SDC setup. This enables us to achieve up to 20-nm lateral and 80-nm axial resolution and image through the entire thickness of mammalian cultured cells using non-fluorogenic DNA-PAINT probes.

## Results

**Measuring achievable resolution using DNA origami structures.** In order to evaluate DNA-PAINT imaging on an SDC, we first designed 2D DNA origami nanostructures[19] with DNA-PAINT extensions at the corners, spaced ~70 and 50 nm apart (Fig. 1c). With our system, we achieved an average localization precision of 12.5 nm calculated by nearest neighbor analysis (NeNA)[20] from single-molecule binding events, translating to a ~29 nm full width at half maximum (FWHM) supported resolution and enabling us to clearly resolve all four corners (Fig. 1c). Next, we imaged DNA origami structures with 12 binding sites arranged in a 4 × 3 grid spaced 20 nm apart (Fig. 1d) with optimized conditions (e.g., higher laser excitation power, see Methods section for further details). Here, we obtained an average localization precision of 7.6 nm, suggesting an achievable FWHM-limited spatial resolution of ~18 nm, allowing us to resolve all 12 grid points spaced 20 nm apart (Fig. 1d). Single binding sites of the 20-nm-grid "sum" image yielded an average localization precision of ~6 nm, in good agreement with the calculated (2D) NeNA localization precision. We then examined our ability to perform three-dimensional super-resolution imaging by placing a cylindrical lens in the detection path, which introduces an optical astigmatism for depth-dependent point-spread-function (PSF) shaping[21, 22] (Fig. 1b. For details on calibration, accuracy, and

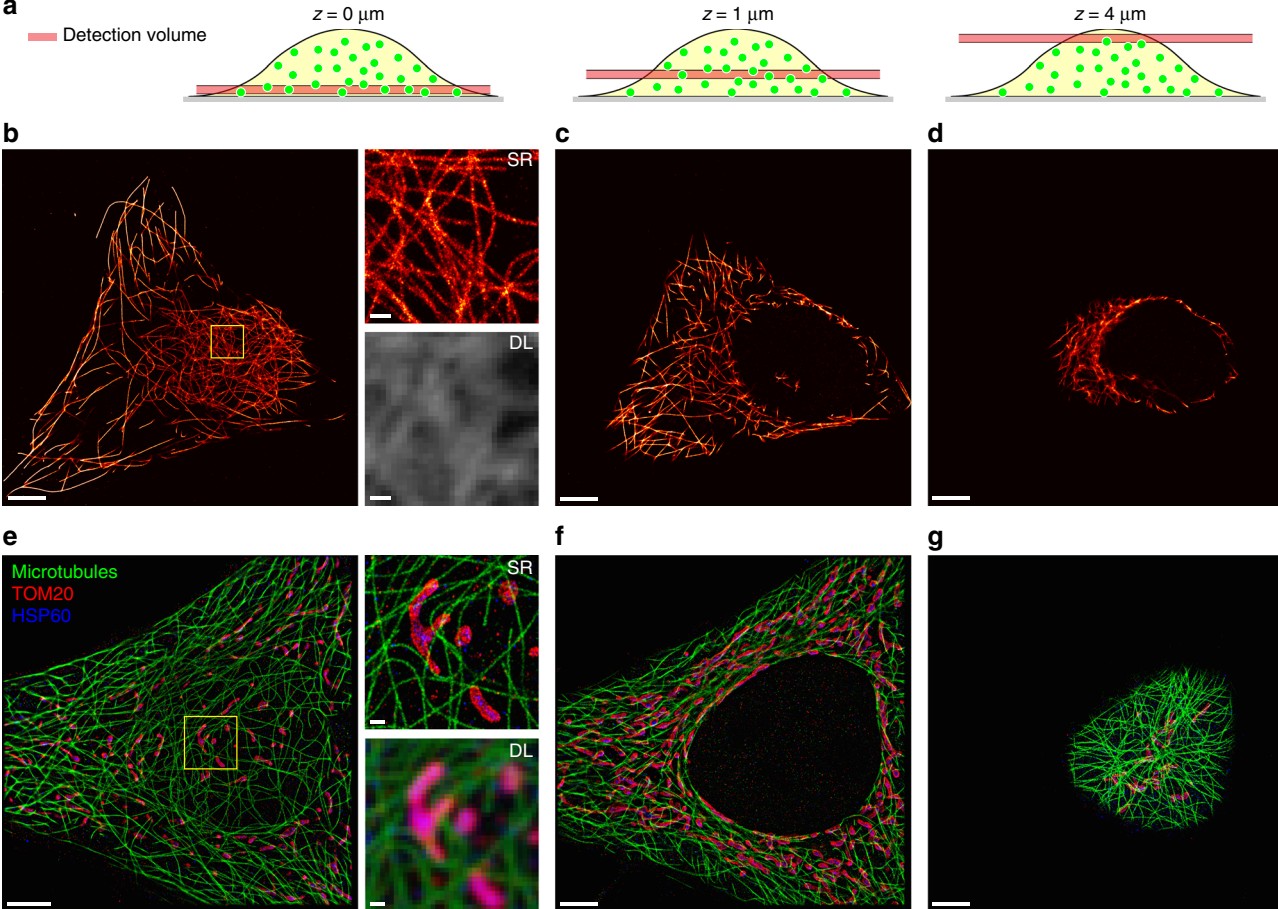

**Fig. 2** Whole-cell DNA- and Exchange-PAINT enabled by SDC microscopy. **a** Schematic drawing of the cell and the detection volume for z positions 0 μm (left), 1 μm (middle), and 4 μm (right). **b** 500-nm slice of the microtubule network in a HeLa cell at the coverslip surface. Left: super-resolved DNA-PAINT image. Right: super-resolved (SR) and diffraction-limited (DL) zoomed-in image of the highlighted area. **c** 500-nm slice of the same cell ~1 μm away from the coverslip surface. **d** 500 nm slice of the same cell ~4 μm away from the coverslip surface. **e–g** Three-target Exchange-PAINT of Alpha-Tubulin (green), TOM20 (red), and HSP60 (blue). **e** Left: DNA-PAINT super-resolution image of a 500-nm slice at the coverslip surface. Right: super-resolved (SR) and diffraction-limited (DL) zoomed-in image of the highlighted area. **f** 500 nm slice of the same cell ~1 μm away from the coverslip surface. **g** 500-nm slice of the same cell ~4 μm away from the coverslip surface. Scale bars, 5 μm (**b–g**), 500 nm (insets in **b** and **e**)

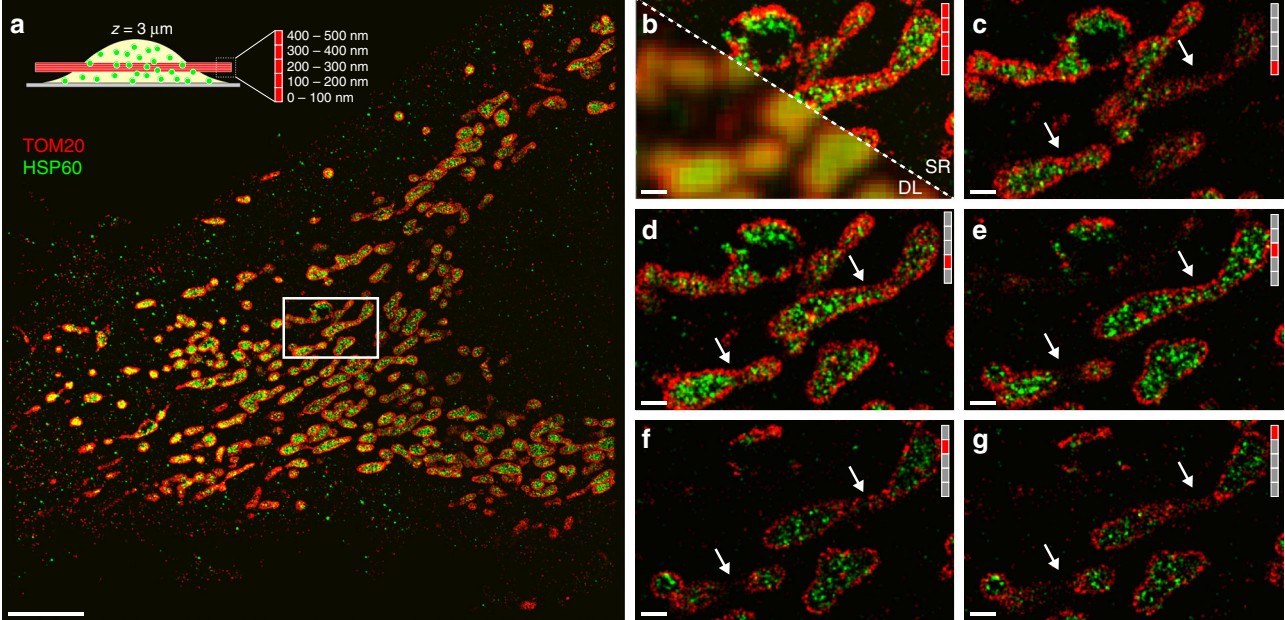

**Fig. 3** Multiplexed 3D-Exchange-PAINT optical section inside a cell. **a** 2D projection of a 500-nm thick SDC slice ~3 μm inside a fixed HeLa cell. Two-target Exchange-PAINT targeting TOM20 (red) and HSP60 (green). Top left: Schematic drawing of the cell along with the detection volume (red). The 500-nm thick detection volume is split up into smaller slices (100 nm thick, see zoom-in) using astigmatism-based 3D super-localization. **b** Comparison of diffraction-limited (DL) and DNA-PAINT image (SR) of the highlighted area in **a**. **c** 3D super-resolved *z*-slice (0–100 nm, see height indicator on the top right) of the highlighted area in **a**. The arrows highlight areas that change appearance throughout the whole 500 nm slice. **d** 3D super-resolved *z*-slice (100–200 nm) of the highlighted area in **a**. **e** 3D super-resolved *z*-slice (200–300 nm) of the highlighted area in **a**. **f** 3D super-resolved *z*-slice (300–400 nm) of the highlighted area in **a**. **g** 3D super-resolved *z*-slice (400–500 nm) of the highlighted area in **a**. Scale bars, 5 μm (**a**), 500 nm (**b**–**g**)

precision performance, as well as detected number of localizations and photons vs. *z*-position in a single depth of field see Supplementary Figs. 1–4). We evaluated this configuration using DNA origami tetrahedron structures[23] displaying DNA-PAINT extensions at each vertex, spaced 100 nm apart. First, we established our ability to resolve this structure in a lateral 2D projection (Fig. 1e). Next, we performed EM-style "averaging" by summing up 42 tetrahedron structures by first aligning them to their center of mass and then implementing a 3D cross-correlation maximization step and observed that we could clearly resolve all four corners in the resulting reconstruction (Fig. 1f–h). In the *xz* projection of the reconstruction, we observed via a cross-sectional histogram a height of 80 nm (Fig. 1i), in good agreement with the designed distance[23] of 82 nm. Additionally, a single vertex of the tetrahedron "sum" image yielded an average localization precision of ~8 nm in *x*, ~8 nm in *y*, and ~19 nm in *z* (Supplementary Fig. 5), in good agreement with the calculated (2D) NeNA localization precision of 7.8 nm.

**Whole-cell 2D imaging of protein targets**. Next, we investigated the feasibility of imaging protein targets in fixed cells. We first coupled DNA-PAINT docking strands covalently to secondary antibodies and performed immunostaining using primary antibodies against alpha-tubulin followed by incubation with DNA-conjugated secondary antibodies[24]. Then we acquired a ~500-nm thick confocal volume, which was subsequently moved with 500 nm steps throughout the whole cell, resulting in 14 imaging planes (total height 6.5 μm) (Fig. 2a–d; Supplementary Fig. 6). We found that we were able to clearly resolve filamentous microtubule structures at the coverslip–cell interface (Fig. 2b), at an intermediate axial position (Fig. 2c; Supplementary Fig. 6), and at the top of the cell (Fig. 2d, for similar measurement with a water immersion objective see Supplementary Fig. 7). Subsequently, we performed three-target Exchange-PAINT[18], sequentially imaging

Alpha-Tubulin, TOM20, and HSP60 proteins and were able to achieve a depth-averaged lateral localization precision of 22 nm based on the NeNA metric of all targets in 11 planes and a total height of 5 μm (Fig. 2e–g; Supplementary Fig. 8). We note that localization precisions calculated according to the Cramér-Rao lower bound (CRLB) of the single-molecule fits[25, 26] (see Supplementary Tables 1 and 2 for details) as well as the achievable precisions according to the NeNA[20] metric are depth-dependent (e.g., worsening with increasing depth), as the number of detected photons decreases with increasing axial depth due to scattering and aberrations (Supplementary Fig. 9).

**3D super-resolution imaging inside cells**. We then used the astigmatism-extended SDC setup to 3D super-resolve TOM20 and HSP60 proteins of mitochondria in a 500-nm thick slice of a HeLa cell, 3 μm away from the coverslip (Fig. 3). A 2D projection of the whole volume (Fig. 3a) demonstrates our ability to visualize TOM20 (red), which localizes to the outer membrane of mitochondria along with HSP60 (green), which is mainly located at the mitochondrial matrix[27]. While TOM20 and HSP60 are clearly resolved in the DNA-PAINT image (Fig. 3a, b), the diffraction-limited representation fails to provide sufficient spatial resolution to discern these features (Fig. 3b). Additionally, astigmatism-based PSF shaping allows each 500-nm *z* section image to be divided into 100-nm thick super-resolution subsections, allowing us to clearly visualize TOM20 localizing to the outer mitochondrial membrane and the matrix localization of HSP60 (Fig. 3c–g; See Supplementary Fig. 10 and Supplementary Movies 1 and 2 for more details as well as Supplementary Movie 3 for examples of RAW frames from image acquisition). We further assayed our achievable resolution at this depth by performing quantitative measurements of the TOM20 localizations in the outer membrane, both axially and laterally. We found that we were able to localize TOM20 proteins with a lateral precision of ~20 nm at

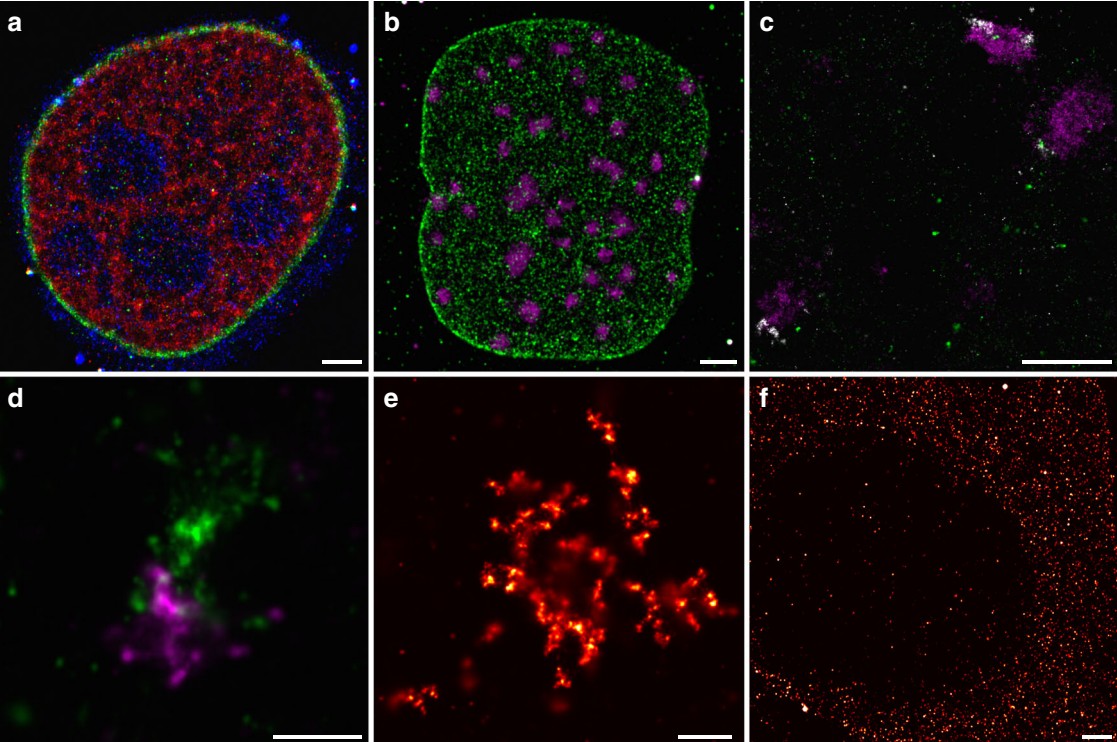

**Fig. 4** DNA-PAINT imaging of distinct types of biomolecular targets at single optical sections. **a** Three-color immunostaining of fixed HeLa cells for histone variant MacroH2A.1 (red), Lamin B (green), and nucleophosmin (blue) at 3.6 µm height from the glass surface. **b** DNA FISH in EY.T4 mouse embryonic fibroblasts for the major satellite (magenta) and immunostaining for Lamin A (green) at 4.6 µm height from the glass surface. **c** Three-color DNA FISH against the major satellite (magenta), minor satellite (white), and telomere regions (green) in mouse embryonic fibroblasts. **d** Two-color DNA FISH against single-copy targets at Xq28 in human female WI-38 fetal lung fibroblast cells. **e** RNA FISH against the Xist RNA in human female IMR-90 fetal lung fibroblast cells. **f** Single-molecule RNA FISH against the CBX5 mRNA in HeLa cells. Scale bars, 2 µm (**a**, **b**, and **f**), 1 µm (**c**–**e**)

vertically oriented membrane sections and with an axial precision of ~40 nm in horizontally oriented membrane sections (see Supplementary Fig. 11 for details).

**Imaging DNA, RNA and protein targets inside cells**. One potential advantage of the SDC platform is the ability to image in crowded environments such as the nucleus, which spans axial heights beyond the typical range of TIRF imaging approaches. Therefore, we next tested the ability of the SDC platform to perform DNA-PAINT imaging on a broad range of biological targets. We performed three-color immunostaining for the nuclear architectural proteins Lamin B and nucleophosmin as well as the histone variant MacroH2A that marks heterochromatin[28] (Fig. 4a) in HeLa cells, as well as separately performing simultaneous immunostaining for Lamin A and fluorescence in situ hybridization (FISH) against the major satellite repeat DNA (Fig. 4b) and three-color DNA FISH against the major satellite, minor satellite, and telomere repeats[29] in mouse embryonic fibroblasts[30] (Fig. 4c). In all cases, we were able to successfully super-resolve each target in the crowded nuclear environment. We next extended our imaging to distinct types of biomolecular targets by performing two-color DNA FISH against single-copy targets[31, 32] (Fig. 4d), RNA FISH against the nuclear Xist long noncoding RNA[33] (Fig. 4e), and single-molecule RNA FISH[34] against the CBX5 messenger RNA (mRNA) in female human fibroblasts (Fig. 4f). Collectively, these results demonstrate the ability of the SDC platform to enable the robust detection of RNA, DNA, and protein targets in a range of sub-cellular environments.

## Discussion

In summary, we have demonstrated that sub-20-nm lateral and sub-80-nm axial resolution using DNA-PAINT in combination with a minimally modified commercial SDC microscope is possible, resulting in roughly a 10-fold improvement over the diffraction limit in both the lateral and axial direction. We show that super-resolution with DNA-PAINT can be achieved in whole cells away from the coverslip for a broad range of biological targets while maintaining a good SNR ratio. Based on our instrumental implementation, we established that 500-nm sectioning is sufficient to provide structural "oversampling", thus enabling continuous and undistorted localizations for whole-cell imaging (Supplementary Figs. 12–14, see also Supplementary Figs. 15–18 for exemplary raw data). Furthermore, experiments, where predefined z-positions have been repeatedly sampled after stage movement show that we can achieve a high degree of reproducibility resulting in resolution errors of only a few nanometers after 10 µm of z-stage travel (Supplementary Fig. 19). We note that we do observe depth-dependent loss in localization precision (and thus resolution) due to a reduction in the number of detectable photons from single-molecule events (Supplementary Fig. 9) as expected by scattering and aberration artifacts. However, we were still able to obtain lateral localization precisions of ~20 nm and axial localization precisions of ~40 nm when imaging TOM20 proteins localizing in the outer membrane of mitochondria, ~3 µm deep inside cells (Supplementary Fig. 11). Some of these aberrations could be corrected for by implementing recent developments in adaptive optics[35].

When compared to HILO[14], SDC-PAINT essentially provides diffraction-limited sectioning capabilities. HILO illumination on the other hand creates micron-sized sheets, which also exhibit a size-dependent axial depth in combination with "uneven" illumination in the field of view. The latter is prevented in the case of our SDC-PAINT implementation by using a beam conditioning unit (BCU) providing a flat-field illumination. While recent advancements in lattice light-sheet[36] and 4Pi microscopy[37] present promising approaches to obtain high SNR through thick specimens (clearly outperforming SDC-PAINT in terms of excitation and detection efficiency and thus achievable spatial resolution), we want to note that they require rather complex hardware, operation, and sample preparation, currently limiting their applicability in standard biology labs. In contrast, SDC microscopes are widely available, straightforward to use and maintain, and are well-suited for the investigation of many biological questions.

We expect SDC-PAINT to enable multiplexed super-resolution imaging of proteins and nucleic acids in a wide range of applications ranging from unraveling chromosome structure in the nucleus using DNA FISH[32] over whole-cell 3D-RNA FISH[34] to imaging of a multitude of organelles and membrane-bound receptors using sequential probe exchange[38].

Further improvement in SDC hardware, e.g., more efficient disks (in terms of light throughput vs. sectioning capability) could further improve the spatial and temporal resolution of SDC-PAINT. In conclusion, the combination of DNA-PAINT and SDC hardware now enables DNA-based multiplexed and quantitative 3D super-resolution imaging in whole cells, which we expect will expand the use of SMLM as an everyday research tool.

## Methods

**Materials.** Unmodified DNA oligonucleotides, fluorescently modified DNA oligonucleotides, and biotinylated DNA oligonucleotides were purchased from MWG Eurofins. M13mp18 scaffold was obtained from New England BioLabs (cat: N4040S). p8064 scaffold for the tetrahedron DNA origami structure was prepared by replacement of the *Bam*HI-*Xba*I segment of M13mp18 by a PCR-amplified fragment of bacteriophage λ DNA, flanked by positions −25 to +25 of the middle of the *Xba*I cut site (TCTAGA or base 6258)[39]. Agarose (cat: 01280.100) was purchased from Biomol. SYBR safe (cat: SS33102) was ordered from Invitrogen. DNA gel loading dye (cat: R0611) was purchased from ThermoFisher. Freeze 'N Squeeze columns (cat: 732-6165) were ordered from Bio-Rad. BSA-Biotin was obtained from Sigma-Aldrich (cat: A8549). Streptavidin was purchased from Invitrogen (cat: S-888). Tris 1 M pH 8.0 (cat: AM9856), EDTA 0.5 M pH 8.0 (cat: AM9261), magnesium 1 M (cat: AM9530G), and sodium chloride 5 M (cat: AM9759) were ordered from Ambion. Ultrapure water (cat: 10977-035) was purchased from Gibco. Potassium chloride (cat: 6781.1) was ordered from Carl Roth. Natrium hydroxide (cat: 31627.290) was purchased from VWR. Tween 20 (cat: P9416-50ML), glycerol (cat: 65516-500 ml), and methanol (cat: 32213-2.5L) were ordered from Sigma. Protocatechuate 3,4-dioxygenase pseudomonas (PCD) (cat: P8279), 3,4-dihydroxybenzoic acid (PCA) (cat: 37580-25G-F), and (+−)-6-hydroxy-2,5,7,8-tetra-methylchromane-2-carboxylic acid (Trolox) (cat: 238813-5 G) were ordered from Sigma. Glass slides (cat: 48811−703) were purchased from VWR. Coverslips were purchased from Marienfeld (cat: 0107032). Epoxy Glue (cat: TC-EPO5-24) was purchased from Toolcraft. Double-sided tape (cat: 665D) was ordered from Scotch.

Tissue culture-treated flasks (cat: 353136) were ordered from Falcon. Dulbecco's Modified Eagle medium (cat: 31966-021), fetal bovine serum (cat: 10500-064), penicillin streptomycin (cat: 15140-122), and nonessential amino acids (cat: 11140-050) were ordered from Gibco. 1× PBS pH 7.2 (cat: 20012-019) and 0.05% Trypsin−EDTA (cat: 25300-054) was purchased from Gibco. Paraformaldehyde (cat: 15710) and glutaraldehyde (cat: 16220) were obtained from Electron Microscopy Sciences. Sodium borohydride > 97% (cat: 4051.1) was purchased from Roth and NH4Cl (cat: 254134-25G) was ordered from Sigma. Bovine serum albumin (cat: A4503-10G) was ordered from Sigma-Aldrich. Normal donkey serum was ordered from Jackson ImmunoResearch (cat: 017-000-121) and Triton X100 (cat: 6683.1) was purchased from Roth. Monoclonal antibodies against alpha-tubulin (used: 1:100) (Thermo Scientific; cat: MA1-80017), HSP60 (used 1:100) (Thermo Scientific; cat: MA3-012), and nucleophosmin (used 1:50) (cat: 32-5200) were purchased from Thermo Scientific. The monoclonal antibodies against MacroH2A (used 1:200) (cat: ab183041) were ordered from Abcam. Polyclonal antibodies against TOM20 (used 1:100) (cat: sc-11415) and Lamin B1 (used 1:100) (cat: sz-6216) were ordered from Santa Cruz. The polyclonal antibodies against

Lamin A (used 1:200) (cat: ab26300) were purchased from Abcam. The secondary antibodies Anti-Rat (cat: 712-005-150) (used in a 1:100 dilution), anti-rabbit (cat: 711-005-152) (used in a 1:100 dilution), and anti-mouse (cat: 705-005-151) (used 1:100) were purchased from Jackson ImmunoResearch. Eppendorf 8-well chambered coverglass (cat: 0030742036) was purchased from Thermo Fisher Scientific. Lab-Tek II (cat: 155409) chambered 8-well coverglass was ordered from Thermo Fisher Scientific. Glass-bottomed 8-well μ-slides (cat: 80827) were purchased from ibidi.

Three buffers were used for sample preparation and imaging: Buffer A (10 mM Tris-HCl pH 7.5, 100 mM NaCl, 0.05% Tween 20, pH 7.5); Buffer B (5 mM Tris-HCl pH 8, 10 mM MgCl2, 1 mM EDTA, 0.05% Tween 20, pH 8); Buffer C (1× PBS pH 8, 500 mM NaCl, pH 8). 100× Trolox: 100 mg Trolox, 430 μl 100% methanol, 345 μl of 1 M NaOH in 3.2 ml H2O. 40× PCA: 154 mg PCA, 10 ml water, and NaOH were mixed and adjusted to pH 9.0. 100× PCD: 9.3 mg PCD, 13.3 ml of buffer (100 mM Tris-HCl pH 8, 50 mM KCl, 1 mM EDTA, 50% glycerol).

**Optical setup.** Results from Figs. 1c–d, 2b–d, Supplementary Figs. 6 and 7 were carried out on an inverted Nikon Eclipse Ti microscope (Nikon Instruments) with the Perfect Focus System, attached to the spinning disk unit Andor Dragonfly. For all experiments except the 2D microtubule stack from Supplementary Fig. 6, an oil-immersion objective (Plan Apo 100×, numerical aperture (NA) 1.45, oil) was used. For the results in Supplementary Fig. 7, a water immersion objective (SR Plan Apo 60×, NA 1.27, water) was used. As excitation laser, a 561 nm (2 W nominal, MPB) was used. The beam was coupled into a multimode fiber going through the Andor Borealis unit reshaping the beam from a Gaussian profile to a homogenous flat top. From there it was passed through the 40 μm pinhole disk. As dichroic mirror a CR-DFLY-DMQD-01 was used. Fluorescence light was spectrally filtered with an emission filter (TR-DFLY-F600-050) and imaged on an electron-multiplying charge-coupled device (EMCCD) camera (iXon 888, Andor Technologies) or a scientific complementary metal oxide semiconductor (sCMOS) camera (Zyla 4.2, 2.0 Andor Technologies). An additional ×1.5 magnification lens was used resulting in a pixel size of 87 nm for the EMCCD and sCMOS (for the latter after 2 × 2 binning). The power at the objective was measured to be ~3% of the input power. As an exemplary calculation, the laser power was set to 1.1 W before fiber coupling. After fiber coupling, ~400 mW was sent into the BCU of the microscope. From there, light was passed through a single-mode fiber and then passed through the disk. The resulting power output at the objective was measured to be 11 mW.

All other experiments were carried out on an inverted Nikon Eclipse Ti microscope (Nikon Instruments) with the Perfect Focus System, attached to a Yokogawa spinning disk unit (CSU-W1, Yokogawa Electric). An oil-immersion objective (Plan Apo 100×, NA 1.45, oil) was used for all experiments. The excitation laser (561 nm, 300 mW nominal, coherent sapphire or 532 nm, 400 mW nominal, Cobolt Samba) was directly coupled into the Yokogawa W1 unit using a lens (focal length *f* = 150 mm). The pinhole size of the disk was 50 μm. As dichroic mirror, a Di01-T405/488/568/647-13 × 15 × 0.5 from Semrock or t540spxxr-uf1 from Chroma was used. Fluorescence light was spectrally filtered with emission filters (607/36 nm from Semrock or ET585/65m + ET542lp from Chroma) and imaged on an EMCCD camera (iXon 897, Andor Technologies), resulting in a pixel size of 160 nm. The power at the objective was measured to be ~10% of the input power. As an exemplary calculation, the laser power was set to 300 mW. The free laser beam was coupled directly into the unit without a fiber or BCU. We measured 30 mW for the resulting power at the objective for this example.

**DNA origami self-assembly.** The tetrahedron DNA origami structures were formed in a one-pot reaction with a 50 μl total volume containing 10 nM scaffold strand (p8064), 100 nM core staples, 100 nM connector staples, 100 nM vertex staples, 100 nM biotin handles, 100 nM DNA-PAINT handles, and 1400 nM biotin anti-handles in folding buffer (1× TE (5 mM Tris, 1 mM EDTA) buffer with 10 mM MgCl2). The solution was annealed using a thermal ramp cooling from 80 to 4 °C over the course of 15 h. After self-assembly, the structures were mixed with 1× loading dye and then purified by agarose gel electrophoresis (1.5% agarose, 0.5× TAE, 10 mM MgCl2, 1× SYBR Safe) at 3 V/cm for 3 h. Gel bands were cut, crushed and filled into a Freeze 'N Squeeze column and spun for 5 min at 1000×*g* at 4 °C.

The 3 × 4 grid motif (spacing 20 nm) and the 4-corner motive of the Rothemund rectangular origami (RRO) were formed in a one-pot reaction with 50 μl total volume containing 10 nM scaffold strand (M13mp18), 100 nM core staples, 1 μM biotinylated staples, and 1 μM DNA-PAINT handles. The folding buffer was 1× TE buffer with 12.5 mM MgCl2. The structures were annealed using a thermal ramp. First, incubating for 5 min at 80 °C, then going from 65 to 4 °C over the course of 3 h. The self-assembled structures were purified as described before[40]. The DNA staple sequences of the tetrahedron DNA origami structures can be found in Supplementary Data 1. As DNA-PAINT docking sites for the tetrahedron structure, the sequence P1 with 10 nucleotides was used (Supplementary Table 3). DNA staple sequences for the RRO structures are listed in Supplementary Data 2. The sequence of the two scaffold strands M13mp18 and p8064 can be found in Supplementary Data 3 and 4. DNA-PAINT docking sites, the DNA-PAINT imagers sequences (P1, 9, nucleotides), and the biotinylated staples sequences (for the RRO) can be found in Supplementary Tables 3 and 4.

**Antibody–DNA conjugates**. The amino groups of secondary antibodies were labeled with thiol-modified DNA oligonucleotides using a hetero bifunctional linker (maleimide-PEG2-succinimidyl ester Sigma) described in earlier work[24]. DNA sequences used for conjugation to different antibodies are listed in Supplementary Table 3.

**Tissue culture**. Cells were grown in a mixture media consisting of 500 ml Dulbecco's Modified Eagle medium, 50 ml fetal bovine serum, 5.5 ml penicillin streptomycin, and 5.5 ml nonessential amino acids. For washing and passaging the cells 1× PBS pH 7.2, and 0.05% Trypsin–EDTA was used.

**Cell seeding**. HeLa (Leibniz Institute DSMZ: Catalogue of Human and Animal Cell Lines (http://www.dsmz.de), cat. no. ACC-57) and BSC1 (ATCC, cat. no. CCL-26) cells were cultured with Eagle's minimum essential medium fortified with 10% FBS with penicillin and streptomycin and were incubated at 37 °C with 5% $CO_2$. At ~30% confluence, cells were seeded into Lab-Tek II chambered coverglass or Eppendorf 8-well chambered coverglass ~24 h before fixation.

**Fiducial marker preparation**. About 40 nm gold nanoparticles used as fiducials for drift correction and channel alignment were diluted 1:25 from the stock bottle (Sigma 753637) in Buffer C and deposited to the sample by centrifugation for 3 min at 500×g.

**DNA FISH**. 3D DNA FISH[32, 41], was performed on Labtek coverglass chamber slides containing EY.T4 transformed mouse embryonic fibroblasts[30] (Fig. 4c) or human WI-38 fetal lung fibroblasts (ATCC CCL-75) (Fig. 4d) at ~70–90% confluence. Samples were rinsed twice in 1× PBS, fixed for 10 min in 1× PBS + 4% (wt/vol) paraformaldehyde for 10 min, and rinse again twice in 1× PBS. Samples were then permeabilized by incubation in 1× PBS + 0.5% (vol/vol) Triton X-100 (Sigma T8787) for 10 min, rinsed in 1× PBS + 0.1% (vol/vol) Tween 20 (Sigma P9416), and further treated for 5 min in 0.1 M HCl. Samples were subsequently rinsed in 2× SSC + 0.1% (vol/vol) Tween 20 (2× SSCT), incubated in 2× SSCT + 50% (vol/vol) formamide for 5 min, then incubated in 2× SSCT + 50% formamide at 60 °C for 60–90 min. A hybridization mix composed of 2× SSCT, 50% formamide, 10% (wt/vol) dextran sulfate, 10 µg RNase A (ThermoFisher EN0531), and oligonucleotide FISH probe at 1.6 µM was then added, samples were denatured for 3 min at 78 °C, and then allowed to hybridize for 36–48 h at 45 (Fig. 4c) or 52 °C (Fig. 4d) on a flatblock thermocycler with a heated lid (Eppendorf 6335000011). Samples were washed four times for 5 min each with 2× SSCT at 60 °C, then twice for 5 min each with 2× SSCT at room temperature. Samples were then stained with 20 ng/µl 4′,6-diamidino-2-phenylindole (DAPI) in 2× SSCT for 5 min, rinsed in 2× SSCT, and then buffer exchanged to Buffer C. Samples were then rinsed in Buffer C, and then transferred to imaging buffer. Slightly lengthened versions of reported oligonucleotide probes targeting the mouse major satellite, minor satellite, and telomere[29] (Fig. 4c) or pools of "Oligopaint" oligonucleotide probes designed using OligoMiner (Fig. 4d) were used (Supplementary Tables 5, 6).

**Single-molecule RNA FISH**. Single-molecule RNA FISH[34] was performed on HeLa cells (ATCC CCL-2) grown to ~70–90% confluency in a Labtek coverglass chamber slide. Samples were rinsed twice in 1× PBS, then fixed in ice-cold 100% methanol at −20 °C for 30 min. A hybridization solution composed of 2× SSCT, 10% (vol/vol) formamide, 10% (wt/vol) dextran sulfate, and "Oligopaint" oligonucleotide FISH probe at 1.6 µM was then added and hybridization was conducted overnight (~16 h) at 37 °C in a humidified chamber placed in a warm room. Samples were then washed twice in 2× SSCT + 10% formamide for 5 min at 37 °C, then once in 2× SSCT for 5 min at 37 °C. DAPI staining and gold nanoparticle deposition was then performed as described for "3D DNA FISH". The Oligopaint probe targeting the CBX5 mRNA consisted of 162 oligo probes (Supplementary Table 6) designed using the "OligoMiner" pipeline[42].

**Xist RNA FISH**. Human IMR-90 fetal lung fibroblasts (ATCC CCL-186) were grown to ~70–90% confluence in a Labtek coverglass chamber slide. Samples were rinsed twice in 1× PBS, then fixed 1× PBS + 4% paraformaldehyde for 10 min, then rinsed in 1× PBS, and then rinsed in 1× PBS + 0.1% (vol/vol) Tween 20. Samples were then rinsed in 2× SSCT and incubated in 2× SSCT + 50% formamide for 5 min. A hybridization solution composed of 2× SSCT, 50% (vol/vol) formamide, 10% (wt/vol) dextran sulfate, and "Oligopaint" oligonucleotide FISH probe at 1.6 µM was then added and hybridization was conducted overnight (~16 h) at 42 °C on flat-block thermocycler with a heated lid. Washing, DAPI staining, and gold nanoparticle deposition was performed as described for "3D DNA FISH". The Oligopaint probe targeting the Xist RNA consisted of 167 oligo probes (Supplementary Table 6) designed using OligoMiner.

**Immunostaining with 4% PFA fixation**. HeLa cells were grown to ~70% confluence in Labtek chambers. Samples were rinsed with 1× PBS, then fixed with 4% paraformaldehyde for 30 min, followed by quenching with 100 mM $NH_4Cl$ for 20 min. After rinsing for 30 min, blocking and permeabilization with 2% BSA + 0.2% Triton in 1x PBS was performed for 30 min. Samples were then incubated with

primary antibodies (antibodies and dilution are described in the Materials section) for at least 1 h and rinsed four times for 10 min in the same buffer. This was followed by a 1-h incubation with oligo-conjugated secondary antibodies. Finally, the sample was rinsed with 1× PBS for 30 min, stained with 20 ng/µl DAPI in PBS for 10 min and rinsed in PBS.

**Combined DNA FISH and immunostaining**. EY.T4 transformed mouse embryonic fibroblasts[30] were grown on glass-bottomed 8-well µ-slides at ~70–90% confluence. DNA FISH protocol was modified with elongation of the PFA fixation step to 30 min, and was completed as described above. After the final wash in 2× SSCT, immunostaining protocol was applied as described above, starting with the step of blocking/permeabilization in 2% BSA + 0.2% Triton X-100 in PBS for 30 min.

**Immunostaining with 3% PFA and 0.1% glutaraldehyde fixation**. HeLa cells were grown to ~70% confluence in Labtek chambers. Samples were rinsed with 1× PBS, and subsequently fixed using 3% paraformaldehyde and 0.1% glutaraldehyde for 12 min. After rinsing twice with 1× PBS, 0.1% $NaBH_4$ was used for quenching for 7 min. After rinsing four times with 1× PBS for 30, 60 s, and twice for 5 min, samples were incubated with 3% BSA and 0.25% Triton X-100 for blocking and permeabilization for 2 h. Afterwards, the cells were incubated with 10 µg/ml of primary antibodies in a solution with 3% BSA and 0.1% Triton X-100 at 4 °C overnight. Next, samples were rinsed again three times for 5 min with 1x PBS. Then, incubation was performed with 10 µg/ml of labeled secondary antibodies in a solution with 3% BSA and 0.1% Triton X-100 at room temperature for 1 h. Finally, cells were rinsed three times for 5 min with PBS.

**Immunostaining with 3% glutaraldehyde fixation**. HeLa cells were grown to ~70% confluence in Labtek chambers. First, the samples were pre-fixed and pre-permeabilized with 0.4% glutaraldehyde and 0.25% Triton X-100 for 90 s. Next, they were quickly rinsed with PBS once followed by fixation with 3% glutaraldehyde for 15 min. Afterwards, samples were rinsed twice (5 min) with PBS and then quenched with 0.1% $NaBH_4$ for 7 min. After rinsing four times with PBS for 30, 60 s, and twice for 5 min, samples were blocked and permeabilized with 3% BSA and 0.25% Triton X-100 for 2 h. Then, samples were incubated with 10 µg/ml of primary antibodies in a solution with 3% BSA and 0.1% Triton X-100 at 4 °C overnight. Cells were rinsed three times (5 min each) with PBS. Then, they were incubated with 10 µg/ml of labeled secondary antibodies in a solution with 3% BSA and 0.1% Triton X-100 at room temperature for 1 h. Finally, samples were rinsed three times with PBS.

**Super-resolution DNA-PAINT imaging with DNA origami**. For chamber preparation, a piece of coverslip (no. 1.5, 18 × 18 mm², ~0.17 mm thick) and a glass slide (3 × 1 inch² 1 mm thick) were sandwiched together by two strips of double-sided tape to form a flow chamber with inner volume of ~20 µl. First, 20 µl of biotin-labeled bovine albumin (1 mg/ml, dissolved in Buffer A) was flown into the chamber and incubated for 2 min. Then the chamber was washed using 40 µl of Buffer A. Second, 20 µl of streptavidin (0.5 mg/ml, dissolved in Buffer A) was then flown through the chamber and incubated for 2 min. Next, the chamber was washed with 40 µl of Buffer A and subsequently with 40 µl of Buffer B. Then ~50 pM of the tetrahedron DNA origami structures were flown into the chamber and allowed to bind for 30 min. Alternatively ~300 pM of the modified RRO structures were incubated for 5 min. Afterwards the chamber was washed with 40 µl of Buffer B again. Finally, the imaging buffer with Buffer B and 1× Trolox, 1× PCA, and 1× PCD with the Cy3b-labeled imager strand was flown into the chamber. For the 4-corner motif on the RRO the imager concentration (P1, 9 nt) was 5 nM, for 3 × 4, 20 nm grid on the RRO the imager concentration (P1, 9 nt) was 1 nM and for the DNA origami tetrahedron 3D measurement the imager concentration (P1, 10 nt) was 2 nM. The chamber was sealed with epoxy before subsequent imaging.

For the 4-corner motif measurement (Fig. 1c) the Andor iXon 888 with a readout bandwidth of 10 MHz at 16 bit and 3x pre-amp gain was used. The electron-multiplying (EM) gain was set to 300. Imaging was performed using the Andor Dragonfly spinning disk unit with an excitation intensity of ~180 W/cm² at 561 nm at the sample (laser was set to ~800 mW before fiber coupling).

For the 3 × 4 grid motif experiment (Fig. 1d) the Andor Zyla 4.2 with a readout bandwidth of 540 MHz at 16 bit was used. Imaging was performed using the Andor Dragonfly spinning disk unit with an excitation intensity of ~247 W/cm² at 561 nm at the sample (laser was set to ~1.1 W before fiber coupling).

For the DNA origami tetrahedron structure experiment (Fig. 1e–i; Supplementary Fig. 5) the Andor iXon 897 with a readout bandwidth of 5 MHz at 16 bit and 5× pre-amp gain was used. The EM gain was set to 100. Imaging was performed using the Yokogawa W1 spinning disk unit with an excitation intensity of ~226 W/cm² at 561 nm at the sample (laser was set to ~38 mW). No additional magnification lens was used resulting in an effective pixel size of 160 nm.

**Super-resolution DNA-PAINT imaging of cells**. The HeLa cells were prepared with the 3% glutaraldehyde fixation protocol and imaged with a sCMOS camera Zyla 4.2 from Andor using the Dragonfly Unit with a pinhole size of 40 µm. For each plane 15,000 frames with a frame rate of 5 Hz and a readout bandwidth of

540 MHz at 16 bit were acquired. $2 \times 2$ binning was used resulting pixel size of 130 nm. The 561 nm excitation laser was set to ~600 mW resulting in an intensity of ~303 W/cm$^2$ at the sample plane. About 14 stacks were acquired with a distance of 500 nm. An imager concentration of 0.7 nM (Cy3b) in Buffer C was used. These methods were used in Fig. 2b–d and Supplementary Fig. 6.

The three-color (microtubules, TOM20, and HSP60) samples were prepared with the 3% PFA and 0.1% glutaraldehyde fixation protocol and imaged with the iXon 897 Ultra camera from Andor using the Yokogawa W1 unit with a pinhole size of 50 μm. For each plane and color 15,000 frames with a frame rate of 5.55 Hz and a readout bandwidth of 3 MHz at 16 bit with a pre-amp gain of 5× were acquired. An EM gain of 200 was used. No additional magnification and no binning was used resulting pixel size of 160 nm. The 561-nm excitation laser was set to ~67 mW resulting in an intensity of ~135 W/cm$^2$ at the sample plane. About 11 stacks were acquired with a distance of 500 nm. As imager (Cy3b) concentration of 5 nM for the microtubules and 4 nM for TOM20 and HSP60 in Buffer C were used. The three colors were acquired sequentially using Exchange-PAINT[40]. These methods were used in Fig. 2e–g, Supplementary Fig. 8, and Supplementary Figs. 12–14.

The two-color (TOM20, HSP60) 3D volume sample was prepared with the 3% PFA and 0.1% glutaraldehyde fixation protocol and imaged with a iXon 897 Ultra camera from Andor using the Yokogawa W1 unit with a pinhole size of 50 μm. For each color 15,000 frames with a frame rate of 5 Hz and a readout bandwidth of 3 MHz at 16 bit with a pre-amp gain of 5× were acquired. An EM gain of 280 was applied. No additional magnification and no binning was used. The resulting pixel size of 160 nm. The 561-nm excitation laser was set to ~70 mW resulting in an intensity of ~417 W/cm$^2$ at sample plane. A planar-convex cylindrical lens with a focal length of 0.5 m ~2 cm in front of the camera was used. The volume was acquired ~3 μm away from the surface. An imager (Cy3b) concentration of 5 nM in Buffer C was used. These methods were used in Fig. 3 and Supplementary Figs. 10 and 11.

The three-color sample in Fig. 4a was imaged with the iXon 897 Ultra camera from Andor using the Yokogawa W1 unit with a pinhole size of 50 μm. About 20,000 (for Nucleophosmin) or 25,000 frames (for Lamin B and Histone) with a frame rate of 5 Hz and a readout bandwidth of 10 MHz at 16 bit with a pre-amp gain of 5x was acquired. An EM gain of 100 was used. No additional magnification and no binning was used. The resulting pixel size was 160 nm using $256 \times 256$ pixel as field of view. An excitation intensity of 50 mW at 561 nm was used, resulting in an effective power density of ~1.2 kW/cm$^2$ at the sample plane. We used Cy3b-imagers with nine nucleotide hybridization length at a concentration of 2 nM was applied in Buffer C (for Lamin B and nucleophosmin) or with 10 nucleotide hybridization length in PBS with addition of 125 mM NaCl (for histone). The three colors were acquired sequentially using Exchange-PAINT[40]. The optical section was acquired at 3.6 μm away from the surface. Average NeNA precision: 32 nm.

The two-color sample shown in Fig. 4b was imaged with the iXon 897 Ultra camera from Andor using the Yokogawa W1 unit with a pinhole size of 50 μm. About 9000–10,000 frames with a frame rate of 5 Hz and a readout bandwidth of 10 MHz at 16 bit with a pre-amp gain of 5× were acquired per color. An EM gain of 100 was used. No additional magnification and no binning was used. The resulting pixel size was 160 nm using $256 \times 256$ pixel as field of view. An excitation intensity of 40 mW at 561 nm was used, resulting in an effective power density of ~950 W/cm$^2$ at the sample plane. We used Cy3b-coupled imager strands with 9 nt hybridization length at 1.5 nM concentration for Lamin A and 1 nM for major satellite in Buffer C. The two colors were acquired sequentially using Exchange-PAINT. The optical section was acquired at 4.6 μm away from the surface. Average NeNA precision: 34 nm.

The three-color sample in Fig. 4c was imaged with the iXon 897 Ultra camera from Andor using the Yokogawa W1 unit with a pinhole size of 50 μm. About 30,000 frames with a frame rate of 5 Hz and a readout bandwidth of 10 MHz at 16 bit with a pre-amp gain of 5× were acquired for all targets. An EM gain of 150 was used. A 2× magnification lens and no binning was used. The resulting pixel size was 80 nm using $512 \times 512$ pixel as field of view. An excitation intensity of 80 mW at 532 nm was used, resulting in an effective power density of ~1.9 kW/cm$^2$ at the sample plane, accounting for the use of a 4× focusing lens. Cy3b-labeled imagers with nine nucleotide hybridization lengths were used at 1 nM in 1× PBS + 125 mM NaCl + PCD/PCA/Trolox (major and minor satellite) or in 1× PBS + 250 mM NaCl + PCD/PCA/Trolox (telomere). The three colors were acquired sequentially using Exchange-PAINT[40]. The optical section was acquired 1.2 μm away from the surface. Average NeNA precision: 10 nm.

The two-color sample shown in Fig. 4d was imaged with the iXon 897 Ultra camera from Andor using the Yokogawa W1 unit with a pinhole size of 50 μm. About 30,000 frames with a frame rate of 5 Hz and a readout bandwidth of 10 MHz at 16 bit with a pre-amp gain of 5× were acquired for each target. An EM gain of 200 was used. No additional magnification and no binning was used. The resulting pixel size was 160 nm using $256 \times 256$ pixel as field of view. An excitation intensity of 50 mW at 561 nm was used, resulting in an effective power density of ~1.2 kW/cm$^2$ at the sample plane, accounting for the use of a 4× focusing lens. Cy3b-labeled imagers with 10 nucleotide hybridization lengths were used at 1 nM in 1× PBS + 125 mM NaCl + PCD/PCA/Trolox. The two colors were acquired sequentially using Exchange-PAINT[40]. The optical sections were acquired 970 (Xq28 probe 1—magenta) and 630 nm (Xq28 probe 2—green) from the surface. Average NeNA precision: 22 nm.

The sample for Fig. 4e was imaged with the iXon 897 Ultra camera from Andor using the Yokogawa W1 unit with a pinhole size of 50 μm. About 20,000 frames with a frame rate of 5 Hz and a readout bandwidth of 10 MHz at 16 bit with a pre-amp gain of 5× were acquired. An EM gain of 200 was used. No additional magnification and no binning was used. The resulting pixel size was 160 nm using $256 \times 256$ pixel as field of view. An excitation intensity of 40 mW at 561 nm was used, resulting in an effective power density of ~1.0 kW/cm$^2$ at the sample plane, accounting for the use of a 4× focusing lens. A Cy3b-labeled imager with a 10 nucleotide hybridization length was used at 2 nM in 1× PBS + 250 mM NaCl + PCD/PCA/Trolox. The optical section was acquired at 450 nm from the surface. Average NeNA precision: 22 nm.

The sample for Fig. 4f was imaged with the iXon 897 Ultra camera from Andor using the Yokogawa W1 unit with a pinhole size of 50 μm. About 20,000 frames with a frame rate of 5 Hz and a readout bandwidth of 10 MHz at 16 bit with a pre-amp gain of 5× were acquired. An EM gain of 200 was used. No additional magnification and no binning was used. The resulting pixel size was 160 nm using $256 \times 256$ pixel as field of view. An excitation intensity of 40 mW at 561 nm was used, resulting in an effective power density of ~1.0 kW/cm$^2$ at the sample plane, accounting for the use of a 4× focusing lens. A Cy3b-labeled imager with a 10 nucleotide hybridization length was used at 2 nM in 1× PBS + 250 mM NaCl + PCD/PCA/Trolox. The optical section was acquired at 370 nm from the surface. Average NeNA precision: 22 nm.

The sample for Supplementary Fig. 7 was prepared with the 3% glutaraldehyde fixation protocol and imaged with the EMCCD camera iXon 888 from Andor using the Dragonfly unit with a pinhole size of 40 μm. For each image 15,000 frames with a frame rate of 5 Hz and a readout bandwidth of 10 MHz at 16 bit with a pre-amp gain of 5× were acquired. An EM gain of 300 was applied. An additional ×1.5 magnification lens was used resulting together with the ×60 objective lens (NA = 1.29, water immersion) in a pixel size of 144 nm. The imaged field of view was $512 \times 512$ pixel. The 561-nm excitation laser was set to ~800 mW resulting in an intensity of ~180 W/cm$^2$ at sample plane. About 13 stacks were acquired with a distance of 500 nm. We used an imager (Cy3b) concentration of 1 nM in Buffer C.

The sample for Supplementary Fig. 19 was prepared with the 3% PFA and 0.1% glutaraldehyde fixation protocol and imaged with an EMCCD camera iXon 897 Ultra from Andor using the Yokogawa W1 unit with a pinhole size of 50 μm. For each image 15,000 frames with a frame rate of 5 Hz and a readout bandwidth of 10 MHz at 16 bit with a pre-amp gain of 5× were acquired. An EM gain of 280 was applied. No additional magnification or binning was used resulting in a pixel size of 160 nm. The imaged field of view was $256 \times 256$ pixel. The 561 nm excitation laser was set to ~60 mW resulting in an intensity of ~357 W/cm$^2$ at sample plane. Five stacks at 0, 2, 5, 2, and 0 μm were acquired. We used an imager (Cy3b) concentration of 5 nM in Buffer C.

**3D DNA-PAINT imaging**. 3D images were acquired using a plan-convex cylindrical lens with a focal length of $f = 0.5$ m, ~2 cm away from the camera chip. The calibration was done as in earlier studies[22]. For the processing of the data the software package Picasso[40] was used.

**Super-resolution data processing**. Super-resolution DNA-PAINT reconstruction, drift correction (Supplementary Fig. 20) and alignment was carried out as described before[40] using the software package Picasso.

**Data availability**. All relevant data are available from the authors upon request.

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

## Acknowledgements

We thank Mingjie Dai, Thomas C Ferrante, Josh Rosenberg, Thomas Schlichthaerle, Joerg Schnitzbauer, and Orsolya K Wade for helpful discussions and technical support. We also thank Kimberly A. Cramer for proofreading the manuscript. This work was supported by the DFG through the Emmy Noether Program (DFG JU 2957/1-1), the SFB 1032 (Nanoagents for the spatiotemporal control of molecular and cellular reactions), the ERC through an ERC Starting Grant (MolMap, Grant agreement number 680241), the Max Planck Society, the Max Planck Foundation, and the Center for Nanoscience (CeNS) to R.J. and NIH 1R01EB018659-01, NIH 1-U01-MH106011-01, ONR N00014-13-1-0593, ONR N00014-14-1-0610 and N00014-16-1-2182, ONR DURIP N00014-16-1-2563, ONR N00014-16-1-2563, NSF CCF-1317291, and Wyss Institute funds to P.Y. B. J.B. was supported by a Damon Runyon Cancer Research Foundation Fellowship. H.M.S. is supported by a Uehara Memorial Foundation Postdoctoral Fellowship. S.K.S. was supported by postdoctoral fellowships from EMBO (ALTF 1278-2015) and Human Frontier Science Program. M.T.S. acknowledges support from the International Max Planck Research School for Molecular and Cellular Life Sciences (IMPRS-LS).

## Author contributions

F.S. designed and performed the experiments, analyzed the data, and wrote the manuscript. J.L.-G. performed cell experiments and wrote the manuscript. B.J.B. designed and performed FISH experiments, analyzed the data, and wrote the manuscript. S.K.S. developed immuno- and FISH-labeling assays, performed experiments, and wrote the manuscript. H.M.S. designed and performed FISH experiments, analyzed the data, and wrote the manuscript. J.B.W. helped with initial designs and experiments. M.T.S. developed software for 3D averaging. H.G. helped with optical calculations. P.Y. and R.J. conceived of and supervised the study, interpreted the data, and wrote the manuscript. All authors reviewed and approved the manuscript.

## Additional information

**Competing interests:** P.Y. and R.J. are co-founders of Ultivue, Inc., a startup company with interest to commercialize DNA-PAINT technology. P.Y. is a co-founder of NuProbe Global. The remaining authors declare no competing financial interests.

