## [Peer Review File · Nature Communications]

Reviewers' comments:

Reviewer #1 (Remarks to the Author):

General

In its beginnings in the mid 1990s, localization microscopy based on optical isolation of fluorescent point targets was conceived to use photostable fluorochromes and their spectral characteristics such as differences in absorption, emission, fluorescence life times etc, in combination with or without stochastic binding to the target regions. The potential of these approaches was overshadowed, however, by the discovery by Lidke (2005), Betzig, Hess and Rust (2006), and in the following many other groups that various procedures of 'photoswitching' offer very efficient ways to realize the required optical isolation even at high densities of fluorescent probes. These Single-Molecule Localization Microscopy (SMLM) methods (e.g. PALM, STORM etc.) still remain quite useful but the difficulties to control the chemical environment in a molecule specific way to optimize the photoswitching process currently have made these approaches an art so far mastered by relatively few laboratories.

In recent years, the Jungmann group has developed a very elegant method called qPAINT to overcome this problem and to achieve high optical and structural resolution even by the application of photostable fluorophores in connection with transient binding to their targets. In the present manuscript, the authors present a DNA-Paint solution which allows to perform high structural resolution localization microscopy with an essentially (apart from the astigmatic lens introduced) conventional Spinning Disk Confocal (SDC) system. Since SDC microscopy is established in a large number of laboratories, in principle neither special hardware nor special single molecule photophysics experience is necessary to perform high optical and structural resolution studies in all those cases where the qPAINT approach using transient binding of short fluorescence labeled oligomers can be applied. This is a great methodological step forwards towards the general implementation of super-resolved fluorescence microscopy into biological research, thus fulfilling the prediction of the Nobel Prize Committee (2014) that these methods will "revolutionize biology and medicine".

In this well written and information dense communication, the authors demonstrate the achievable resolution with two- and three-dimensional DNA origami structures and show the general applicability by imaging a large variety of different cellular targets including proteins, DNA and RNA deep in cells, even inside cell nuclei.

A basic problem of switching based localization microscopy for multiplexing of targets is the requirement that for many fluorophores, the chemical environment has to be adapted appropriately. The authors of this ms avoid this complication by typically applying the same

imager (Cy3b) in combination with a recently developed approach called Exchange-Paint. Another major advantage of their method is that in many cases, the required illumination intensities could be drastically reduced to values below 1 kW/cm². This enables the use of much more economic laser sources, or much larger fields of view, respectively.

Specific remarks

1) First 'proof-of-principle' localization microscopy experiments using photostable dyes in connection with confocal laser scanning fluorescence microscopy and applied to study cellular nanostructures have already been performed in the beginning of the 2000s by a German-Israeli group (Esa et al. J. Microscopy); the optical resolution in the target region was enhanced to few tens of nm in 3D.

2) Figures 2,3. Some quantitative characterization of the images should be highly useful for comparison with other localization microscopy approaches, such as number of single molecules/ μm^2 , histogram of localization precision, histogram of distance distribution etc.

3) Methods: "The power at the objective lens was about 1% of the power of the laser". Any damage of the objective lenses, or the dichroic mirrors?

4) Methods: "As excitation laser, a 561 (2 W nominal...was used)". Compared to photoswitching based widefield localization microscopy or to conventional confocal microscopy, this laser power is about ten times higher. In other cases, lower laser powers were needed but even 400 mW appears to be still quite high. Perspectives to further reduce the laser power required?

5) Methods: "For each plane and color 15,000 frames with a frame rate of 5.55 Hz ... were acquired". For 3 colors this would require a total acquisition time of about 2,5 hours. This is a pretty long time to maintain the high mechanical stability required or at least useful to reduce then calibration effort.

6) The authors abstain from any remark about the biological significance of the beautifully resolved biological images shown. In a revised version, at least some review articles might be cited concerning biological application perspectives.

Reviewer #2 (Remarks to the Author):

The authors describe the combination of spinning disk confocal microscopy with exchange-PAINT, and present a number of nice application examples. Whilst it could be argued that this is a fairly obvious combination, it has not (to my knowledge) been demonstrated previously - no

doubt others (myself included) have been put off due to the fairly mediocre results obtained when combining spinning disk confocal with PALM/STORM. The higher photon counts available from PAINTE probes, however, would seem to make the difference between a technical nicety, and a truly viable technique.

In this paper the authors have put a lot of effort into performing spinning-disk based PAINTE on a broad range of different structures, and demonstrating conclusively that this combination has great potential. I don't doubt that this paper will prove a useful resource and inspiration to many in the field.

Now for my concerns:

Resolution (particularly axial)

=====

The 15x50nm resolution claim is based on the tetrahedral DNA origami structure, yet the paper is written as though this will apply equally to the 3D stacks. The axial localization precision is known to degrade rapidly with both background (which is likely to be significantly higher in the cell samples) and with increasing spherical aberration. No attempt is made to quantify axial resolution at depth in the sample, despite having imaged structures (microtubules & nuclear lamin) from which it should be reasonably easy to extract some form of axial resolution metric (at the claimed resolution, even the tops of mitochondria should be useable).

Axial aberrations

=====

This is somewhat related to the above. The manuscript seems to present top down views, or serial sections, avoiding x-z views. It would be nice to see some x-z views, particularly for the mitochondria, which are expected to be a nice round tube. In the supplementary movie, many of the mitochondria seem to be 'open' on the top and bottom. Another good visual indication of the presence / absence of axial aberrations is x-z views of microtubules, showing that they line up from one section to the next.

Even if (as I suspect) the axial resolution is not (nearly) as good as the 50nm claim within the extended samples, I don't think it a critical flaw in the method - It is extremely difficult to get well behaved z-localization in thicker samples, especially with astigmatism and single objective systems, and the images are already pretty compelling. My real concern is more that the authors seem to be hiding this aspect / being deliberately optimistic about the realistically achievable z behaviour. This is problematic as they are trying to sell the method as being almost as good as interferometric 3D, and I think this is an extremely bold claim. That is not to say that there aren't

many applications where I'd choose spinning disk PAINT over, e.g. 4Pi-SMS or iPALM, but a more balanced treatment of the limitations would be much more valuable to the community (especially to new users).

Reviewer #3 (Remarks to the Author):

In this manuscript, Schueder et al. demonstrate high-density astigmatic 3D super-resolution microscopy in conjunction with spinning disk confocal (SDC) illumination and detection, facilitated by DNA-PAINT labelling. This work builds upon previous systems combining single-molecule localisation and SDC, differentiating itself with the addition of a cylindrical lens in the image path and by the use of PAINT labelling.

The use of SDC reduces signal background by limiting the detection volume of the microscope to a ~500nm thick axial plane. However, sample illumination is not limited and out-of-focus fluorophores are still excited and bleached, effectively reducing the labelling efficiency in traditional fluorescence labelling methods such as PALM and (d)STORM. In order to achieve a high localisation density, Schueder et al. implement DNA-PAINT labelling in which each target can theoretically be imaged indefinitely.

It is this reviewer's opinion that this work is somewhat convincing but significant additional quantification and evaluation of the system is needed before it is of an acceptable level to be published in Nature Communications. In its current state, the manuscript does not fairly address (or mention) the limitations of the instrumentation or provide a comparison to other state of the art methodologies.

First and foremost I would like to see SI of the raw single-molecule fluorescence data, SDC is well known that to severely limit the total detected photons due to inefficiencies when compared to other widefield techniques, while the bright fluorophores used in DNA PAINT somewhat reduce get around this, quantification (via a histogram) of the total detected photons of the raw data should be included.

Quantification and discussion should also be provided for:

1. How the pinholes of SDC affects single-molecule detection. How does the position of the fluorophore relative to the confocal volume affect the detected astigmatic PSF or its intensity and are there associated localisation artefacts?
 - a. Fluorescent beads, or some other non-bleaching and diffraction-limited source, should be scanned through the detection volume laterally and axially in order to quantify these issues.
 - b. How well do the localisations obtained in these experiments match the known position of the stage?
 - c. Raw images of representative data collection should be shown. Individual fluorescent events should be highlighted at a range of axial and lateral positions within confocal volumes.
2. Detection efficiency across depth of field and as a result localisation density across multiple

image planes

- a. Are successive image planes overlapped so as to maintain a constant localisation density across the sample?
- b. How does the localisation density vary across the depth of field?
3. Measured localisation precision without summation of multiple structures.
 - a. The true localisation precision of the instruments should be measured by imaging static diffraction-limited fluorescent beads at a range of axial depths.
 - b. A method such as that used in Gahlmann et al. (Nano Lett 2013) should be used to produce plots of lateral and axial localisation precision vs detected photons. This should also be done for beads at the periphery of the detection volume.
4. How axial depth into sample affects imaging.
 - a. How does the localisation density change as depth into the sample increases?
 - b. Is the number of detected photons per localisation dependant on axial depth into sample?
 - c. Is there any lateral correction required to stitch together localisations from image planes across the sample?
5. The drift correction methods require more detail quantification. It is stated in the Methods section that 40 nm gold nanoparticles were used for correction with Picasso software.
 - a. What is the measured drift of the sample across the typical timescales used in the image acquisition?
 - b. What is the precision of temporal cross-correlation methods used by Picasso?
 - c. How are fiducial beads distributed within the sample and how are they imaged?
6. In Figure S3, the cross-correlation of data acquired from the same axial plane within a sample is used to evaluate position reproducibility. How do the quoted coefficients of 0.93 and 0.95 relate to real space errors? This should be quoted in nanometers.

In addition to these points, a comparison to state of the art techniques capable of imaging deep into tissue should be included in the Discussion section. This should include localisation techniques such as astigmatism, biplane, double-helix point-spread-function and multi-focal microscopy when used with selective plane illumination capable of imaging to a similar depth as presented in this work (HILO and light-sheet variations) as well as lattice light sheet. The discussion should include aspects such as signal to background level, depth of field, applicability to labelling methods, acquisition speed and ease of use.

Major recommendations to current manuscript:

1. It is unfair to compare TIRF and HILO illumination with the same statement of 'only offering limited sample penetration depths' in the first paragraph of the introduction. TIRF illumination is severely limited to depths of ~500nm whereas HILO is capable of facilitating 3D single-molecule imaging at depths >10 μ m without the requirement for PAINT labelling methods. Please clarify this statement with appropriate data or reference.
2. In the second paragraph of the introduction, the discussion of reduced excitation intensity and ability to switch dyes into the dark state is only applicable to STORM dyes and not PALM dyes. SDC illumination is not ideal for PALM imaging, but this is due to bleaching of molecules out of

the focal plane, resulting in a large fraction of ‘missed’ localisations. This point should be clarified and extended to include limitations of the SDC system for imaging PALM dyes.

3. In the first paragraph of the Discussion section, it is stated that ‘imaging performance is maintained over the full thickness of a mammalian cell’ although no metrics of performance have been evaluated as a function of axial depth into the sample. As mentioned in the above section, this analysis must be present in order for such claims to be made. Further quantification is needed to support this statement or it should be removed entirely.

4. In figure 1, errors should be quoted and explained with the measurements presented in d and i.

Minor recommendations to current manuscript:

1. ‘TIRF’ is a more accepted acronym compared to ‘TIR’ please change ‘TIR’ to ‘TIRF’ throughout the text.

2. At certain places throughout the text the vocabulary points to a biased position on the subject. Examples of this include but are not limited to:

a. The reference to PAINT techniques as ‘easy-to-implement’ in the introduction, as this is not necessarily the case in cell imaging.

b. The reference to SDC as a ‘promising alternative’ to selective plane illumination in the introduction.

c. The reference to SDC detection as ‘Essentially wide-field’ in the introduction.

d. ‘Encouraged by this result’ in the Results section.

e. ‘in excellent agreement’ in the Results section.

f. ‘Buoyed by our success’ in the Results section.

g. ‘demonstrating exquisite optical sectioning’ in the Results section.

h. ‘lattice light-sheet and 4Pi microscopy present excellent approaches’ in the Discussion section.

i. ‘making it a truly ‘all-in-one’ instrument’ in the Discussion section.

Please change any biased or leading terminology to be more impartial, with special emphasis on examples f, g and i.

Reviewers' comments:

Reviewer #1 (Remarks to the Author):

General

In its beginnings in the mid 1990s, localization microscopy based on optical isolation of fluorescent point targets was conceived to use photostable fluorochromes and their spectral characteristics such as differences in absorption, emission, fluorescence life times etc, in combination with or without stochastic binding to the target regions. The potential of these approaches was overshadowed, however, by the discovery by Lidke (2005), Betzig, Hess and Rust (2006), and in the following many other groups that various procedures of 'photoswitching' offer very efficient ways to realize the required optical isolation even at high densities of fluorescent probes. These Single-Molecule Localization Microscopy (SMLM) methods (e.g. PALM, STORM etc.) still remain quite useful but the difficulties to control the chemical environment in a molecule specific way to optimize the photoswitching process currently have made these approaches an art so far mastered by relatively few laboratories.

In recent years, the Jungmann group has developed a very elegant method called qPAINT to overcome this problem and to achieve high optical and structural resolution even by the application of photostable fluorophores in connection with transient binding to their targets.

In the present manuscript, the authors present a DNA-Paint solution which allows to perform high structural resolution localization microscopy with an essentially (apart from the astigmatic lens introduced) conventional Spinning Disk Confocal (SDC) system. Since SDC microscopy is established in a large number of laboratories, in principle neither special hardware nor special single molecule photophysics experience is necessary to perform high optical and structural resolution studies in all those cases where the qPAINT approach using transient binding of short fluorescence labeled oligomers can be applied. This is a great methodological step forwards towards the general implementation of super-resolved fluorescence microscopy into biological research, thus fulfilling the prediction of the Nobel Prize Committee (2014) that these methods will "revolutionize biology and medicine".

In this well written and information dense communication, the authors demonstrate the achievable resolution with two- and three-dimensional DNA origami structures and show the general applicability by imaging a large variety of different cellular targets including proteins, DNA and RNA deep in cells, even inside cell nuclei.

A basic problem of switching based localization microscopy for multiplexing of targets is the requirement that for many fluorophores, the chemical environment has to be adapted appropriately. The authors of this ms avoid this complication by typically applying the same imager (Cy3b) in combination with a recently developed approach called Exchange-Paint.

Another major advantage of their method is that in many cases, the required illumination intensities could be drastically reduced to values below 1 kW/cm². This enables the use of much more economic laser sources, or much larger fields of view, respectively.

We are grateful to the positive evaluation of our manuscript by this reviewer and the encouraging comments regarding the implementation of DNA-PAINT on a Spinning Disk Confocal microscope.

Specific remarks

1) First 'proof-of-principle' localization microscopy experiments using photostable dyes in connection with confocal laser scanning fluorescence microscopy and applied to study cellular nanostructures have already been performed in the beginning of the 2000s by a German-Israeli group (Esa et al. J. Microscopy); the optical resolution in the target region was enhanced to few tens of nm in 3D.

We thank the reviewer for this comment. We now cite the reference in the introduction to super-resolution.

2) Figures 2,3. Some quantitative characterization of the images should be highly useful for comparison with other localization microscopy approaches, such as number of single molecules/ μm^2 , histogram of localization precision, histogram of distance distribution etc.

This is a very useful suggestion and we apologize for not having included such metrics in greater detail in the initial submission. We now have added more quantification such as localization precision, number of detected photons, and distance measurements throughout the main text and the supplementary information.

3) Methods: "The power at the objective lens was about 1% of the power of the laser". Any damage of the objective lenses, or the dichroic mirrors?

We thank the reviewer for pointing us to this potentially confusing statement. We have now rephrased the sentence and added more specific details in the materials and methods section, stating laser power values at various positions in the excitation light path. See also answer to point 4 below.

4) Methods: "As excitation laser, a 561 (2 W nominal...was used)". Compared to photoswitching based widefield localization microscopy or to conventional confocal microscopy, this laser power is about ten times higher. In other cases, lower laser powers were needed but even 400 mW appears to be still quite high. Perspectives to further reduce the laser power required?

Further to the discussion in point (3) above, we have now rephrased the sentences in the materials and methods section and added more details about excitation intensities used in this study. As an

example for the Andor Dragonfly SDC unit, we used a 561 nm laser source (2 W nominal), which was set to 1.1 W before fiber coupling for the 20-nm-grid imaging experiment. After fiber coupling, ~400 mW was sent into the Beam Conditioning Unit (BCU) of the microscope. From there, it was passed through a single-mode fiber and then passed through the disk. The resulting power output at the objective was measured to be 11 mW.

For an example case of the Yokogawa W1 unit, the free laser beam was coupled directly into the unit without a fiber or BCU. Here, 300 mW laser power was measured before entering the microscope, and 30 mW was the resulting power at the objective.

These values lead now to ~3 % intensity at the objective when compared to the input power for the Dragonfly unit and ~10 % for the Yokogawa unit.

To further decrease the necessary excitation power, one could e.g. think about using a smaller spacing of pinholes, thus achieving a higher power throughput at the cost of potentially increasing crosstalk between the pinholes, ultimately limiting the sectioning capability of the spinning disk.

Further technical improvements in upcoming SDC microscopes could lead to more efficient transmission properties of the disks. We have added a sentence in the discussion where we mention this.

- 5) Methods: “For each plane and color 15,000 frames with a frame rate of 5.55 Hz ... were acquired”. For 3 colors this would require a total acquisition time of about 2,5 hours. This is a pretty long time to maintain the high mechanical stability required or at least useful to reduce the calibration effort.

We agree with the reviewer that faster image acquisition times would be ideal, also in terms of improving the experimental throughput. However, it is not a limiting factor in terms of drift stability for our measurements: We’ve employed a focus stabilization system (Nikon Perfect Focus), which allows us to maintain a constant z-focus over the course of the measurement. No further axial drift correction was necessary. Any xy-drift was corrected using image cross-correlation and/or fiducial markers. We have now included quantitative measurements of our drift correction performance in the supplementary information (Supplementary Figure 20).

- 6) The authors abstain from any remark about the biological significance of the beautifully resolved biological images shown. In a revised version, at least some review articles might be cited concerning biological application perspectives.

We thank the reviewer for raising this important point. We have now added a sentence in the discussion section together with references concerning potential biological applications such as co-localization studies of nuclear DNA and proteins, whole-cell 3D RNA FISH and potentially tissue imaging.

Reviewer #2 (Remarks to the Author):

The authors describe the combination of spinning disk confocal microscopy with exchange-PAINT, and present a number of nice application examples. Whilst it could be argued that this is a fairly obvious combination, it has not (to my knowledge) been demonstrated previously - no doubt others (myself included) have been put off due to the fairly mediocre results obtained when combining spinning disk confocal with PALM/STORM. The higher photon counts available from PAINT probes, however, would seem to make the difference between a technical nicety, and a truly viable technique.

In this paper the authors have put a lot of effort into performing spinning-disk based PAINT on a broad range of different structures, and demonstrating conclusively that this combination has great potential. I don't doubt that this paper will prove a useful resource and inspiration to many in the field.

We thank the reviewer for the positive evaluation of our work.

Now for my concerns:

Resolution (particularly axial)

=====

The 15x50nm resolution claim is based on the tetrahedral DNA origami structure, yet the paper is written as though this will apply equally to the 3D stacks. The axial localization precision is known to degrade rapidly with both background (which is likely to be significantly higher in the cell samples) and with increasing spherical aberration. No attempt is made to quantify axial resolution at depth in the sample, despite having imaged structures (microtubules & nuclear lamin) from which it should be reasonably easy to extract some form of axial resolution metric (at the claimed resolution, even the tops of mitochondria should be useable).

We thank the reviewer for raising this point. We have now added a more detailed characterization of localization precisions and distance measurements to the Supplementary Information of the manuscript. We have added a table with the localization precision values for the lateral resolution of all image planes of Figure 2. We further included more detailed characterization in the form of histograms of detected photons per plane for Figure 2 as well. Finally, as suggested by the reviewer, we have

added a lateral and axial metric for distance measurements at 3 μm depth from the dataset presented in Figure 3. We have performed a cross-sectional histogram analysis of a yz-projection of TOM20 in single mitochondria and now report these values in Supplementary Figure 11.

We agree with the reviewer that both axial and lateral localization precision is depth-dependent and will deteriorate with increasing depth due to the reduced amount of detected photons for deeper sample penetration due to scattering and other aberration artifacts.

We apologize for the imprecise statement regarding the achievable lateral and axial resolution through whole cells. We have changed the wording in the manuscript to reflect this and also added a section in the main text and discussion.

Axial aberrations

=====

This is somewhat related to the above. The manuscript seems to present top down views, or serial sections, avoiding x-z views. It would be nice to see some x-z views, particularly for the mitochondria, which are expected to be a nice round tube. In the supplementary movie, many of the mitochondria seem to be 'open' on the top and bottom. Another good visual indication of the presence / absence of axial aberrations is x-z views of microtubules, showing that they line up from one section to the next.

We thank the reviewer for this valuable suggestion and have now added an xy-view of single mitochondria in the Supplementary Information (same SI figure 11 as discussed in your point above). Furthermore, we have added additional visualization by scanning through xz-slices of the TOM20 dataset in Supplementary Movie 2. We would like to note, that we indeed expect to see most of the mitochondria in the single slice in Figure 3 with 'open' top and/or bottom, as our resolvable z-range is ~ 600 nm. However, one can identify mitochondria in the video slices with smaller diameter, where indeed top and bottom are 'closed'. We would also like to point out, that we don't employ optical astigmatism for z-super-resolution for the optical sections in Figure 2 and 4. However, we have added more results concerning the absence of aberrations between different optical sections in Supplementary Figures 12–14, where continuous microtubules from consecutive sections can be seen lining up.

Even if (as I suspect) the axial resolution is not (nearly) as good as the 50nm claim within the extended samples, I don't think it a critical flaw in the method - It is extremely difficult to get well behaved z-localization in thicker samples, especially with astigmatism and single objective systems, and the images are already pretty compelling. My real concern is more that the authors seem to be hiding this aspect / being deliberately optimistic about the realistically achievable z behaviour. This is problematic as they are trying to sell the

method as being almost as good as interferometric 3D, and I think this is an extremely bold claim. That is not to say that there aren't many applications where I'd choose spinning disk PAINT over, e.g. 4Pi-SMS or iPALM, but a more balanced treatment of the limitations would be much more valuable to the community (especially to new users).

We thank the reviewer for raising this point. We agree that the achievable spatial resolution is dependent on the depth in the sample (see also newly added Supplementary Figure 9 for number of detected photons and Supplementary Table 1 and 2 for calculated localization precisions at different depths in the sample). We do apologize if we created the impression that DNA-PAINT on a Spinning Disk System could achieve resolution comparable to e.g. iPALM or 4Pi-SMS, which is definitely not the case. It was not our intention to make a competing statement with the interferometric 3D approaches. We have revised the manuscript to clarify this and added now more detail in the discussion, comparing SDC-PAINT with higher-resolution methods such as iPALM, 4Pi-SMS, and LLS. While SDC-PAINT does not achieve as high localization precision and thus resolution as aforementioned methods, it is considerably easier to implement in terms of instrumentation and complexity of sample preparation.

Again we thank the reviewer for raising these concern, which we hope to have clarified in the revised version of the manuscript.

Reviewer #3 (Remarks to the Author):

In this manuscript, Schueder et al. demonstrate high-density astigmatic 3D super-resolution microscopy in conjunction with spinning disk confocal (SDC) illumination and detection, facilitated by DNA-PAINT labelling. This work builds upon previous systems combining single-molecule localisation and SDC, differentiating itself with the addition of a cylindrical lens in the image path and by the use of PAINT labelling.

The use of SDC reduces signal background by limiting the detection volume of the microscope to a ~500nm thick axial plane. However, sample illumination is not limited and out-of-focus fluorophores are still excited and bleached, effectively reducing the labelling efficiency in traditional fluorescence labelling methods such as PALM and (d)STORM. In order to achieve a high localisation density, Schueder et al. implement DNA-PAINT labelling in which each target can theoretically be imaged indefinitely.

It is this reviewer's opinion that this work is somewhat convincing but significant additional quantification and evaluation of the system is needed before it is of an acceptable level to be published in Nature

Communications. In its current state, the manuscript does not fairly address (or mention) the limitations of the instrumentation or provide a comparison to other state of the art methodologies.

First and foremost I would like to say that of the raw single-molecule fluorescence data, SDC is well known that to severely limit the total detected photons due to inefficiencies when compared to other widefield techniques, while the bright fluorophores used in DNA PAINT somewhat reduce get around this, quantification (via a histogram) of the total detected photons of the raw data should be included.

We thank the reviewer for the favorable review of our work. We appreciate the suggestion to present some of the RAW data and include representative examples in Supplementary Figures 15–18. We also include more extensive quantification of detected photons and obtainable localization precisions, as suggested (see also later points).

Quantification and discussion should also be provided for:

1. How the pinholes of SDC affects single-molecule detection. How does the position of the fluorophore relative to the confocal volume affect the detected astigmatic PSF or its intensity and are there associated localisation artefacts?
 - a. Fluorescent beads, or some other non-bleaching and diffraction-limited source, should be scanned through the detection volume laterally and axially in order to quantify these issues.

We now include a quantitative measurement of the number of localizations detected vs. the axial position as well as the number of detected photons vs. the axial position in a single optical section in Supplementary Figure 3.

- b. How well do the localisations obtained in these experiments match the known position of the stage?

We now include a more quantitative description of estimated z-position vs. stage position, a histogram of the deviation to the true position for all localizations, and z-accuracy measurements at different axial positions from calibration data in Supplementary Figure 2. We furthermore compare SDC data to TIRF data with this metric in the same figure.

- c. Raw images of representative data collection should be shown. Individual fluorescent events should be highlighted at a range of axial and lateral positions within confocal volumes.

We now include exemplary snapshots of RAW images in the supplementary information in addition to intensity vs. time traces in Supplementary Figures 15–18.

2. Detection efficiency across depth of field and as a result localisation density across multiple image planes
 - a. Are successive image planes overlapped so as to maintain a constant localisation density across the sample?

Yes they do overlap. We have now added a more detailed characterization of this fact in Supplementary Figures 12–14, in which we compare consecutive image planes from the microtubule data in Figure 2e-g, to highlight overlapping image planes.

- b. How does the localisation density vary across the depth of field?

We have now added Supplementary Figure S3 which presents data for the number of localizations vs. the axial position of a bead sample, which shows that the density is relatively constant and only falls off towards the edges of the detection volume.

3. Measured localisation precision without summation of multiple structures.
 - a. The true localisation precision of the instruments should be measured by imaging static diffraction-limited fluorescent beads at a range of axial depths.

We thank the reviewer for this suggestion. We now report localization precisions and nearest neighbor distance measurements from single-fluorophore detection events for datasets in Figure 1, 2 and 3 (see Supplementary Table 1 and 2 as well as main text description and discussion).

*In addition we think that our evaluation of x, y, and z localization precision from sum images presented in Supplementary Figure 5 should present a fair estimation of achievable image resolution, as it allows the characterization – by nature of the analysis method – of an ‘average’ localization precision (and downstream calculated localization-precision-limited spatial resolution). We also note that aligning (and thus summing up) single-molecule localization events by their center of mass was used in other studies to assess the achievable localization precision of a microscope [See e.g. Xu, K., Babcock, H. P., & Zhuang, X. (2012). Dual-objective STORM reveals three-dimensional filament organization in the actin cytoskeleton. *Nature Methods*, 1–6. <http://doi.org/10.1038/nmeth.1841>].*

- b. A method such as that used in Gahlmann et al. (Nano Lett 2013) should be used to produce plots of lateral and axial localisation precision vs detected photons. This should also be done for beads at the periphery of the detection volume.

We thank the reviewer for this suggestion and have added such plot in Supplementary Figure S4.

4. How axial depth into sample affects imaging.
 - a. How does the localisation density change as depth into the sample increases?

We have evaluated the localization density at four axial positions in the dataset from Figure 2. The results are now included in Supplementary Figure S9 in the bottom right graph. In brief, we analyzed 10 random round pick areas with diameter of 320 nm along microtubules for axial depths $z = 0, 1500, 3500,$ and 4500 nm, respectively. We only observe a slight positional dependency, which falls within measurement error. In conclusion, the number of detected localizations per unit area remains relatively constant for increasing z -depth. However, we note that this will most likely drop for deeper z -penetration due to more severe aberration and scattering effects, which cannot be compensated by our adaptive thresholding algorithm used to detect single-molecule events in raw datasets.

- b. Is the number of detected photons per localisation dependant on axial depth into sample?

Yes, the number of detected photons per localization is dependent on the axial depth in the sample. We have added a detailed analysis of detected photons at different axial positions in Supplementary Figure 9. Generally speaking, the number of detected photons decreases with increasing depth, as expected due to effects such as scattering, refractive index mismatch and other aberrations. We also have revised the main text of the manuscript, discussing these effects and also mention possibilities how this effect might be reduced e.g. by the use of adaptive optics.

- c. Is there any lateral correction required to stitch together localisations from image planes across the sample?

We thank the reviewer for pointing us to this important point. We have added a detailed study regarding this question in Supplementary Figures 12–14. We investigate the potential effect of lateral aberrations for consecutive image planes and show that no correction is necessary. Part of the localizations from adjacent planes do overlap and images can be merged in a continuous fashion.

5. The drift correction methods require more detail quantification. It is stated in the Methods section that 40 nm gold nanoparticles were used for correction with Picasso software.

- a. What is the measured drift of the sample across the typical timescales used in the image acquisition?

We do apologize for not including more details about the drift correction method. We have now added a typical xy -drift trace for a dataset from the paper in Supplementary Figure 20. (See also response to point (b) below)

b. What is the precision of temporal cross-correlation methods used by Picasso?

We have now added a characterization of x- and y- localization precision of a gold bead post-drift-correction with the RCC method in Supplementary Figure 20.

c. How are fiducial beads distributed within the sample and how are they imaged?

The method sections contains the following: "For the nanoparticles, we added 200-500 μ l of a 1:25:1-50 dilution of 40 nm Au particles from the stock in Buffer C to the well of the chamber, spun for 3' at 500 RCF (i.e. 500 x g), then rinsed once in Buffer C."

The nanoparticles are imaged simultaneously with DNA-PAINT image acquisition in the same channel.

6. In Figure S3, the cross-correlation of data acquired from the same axial plane within a sample is used to evaluate position reproducibility. How do the quoted coefficients of 0.93 and 0.95 relate to real space errors? This should be quoted in nanometers.

We thank the reviewer for pointing us to this issue. We have added a cross-sectional analysis of two adjacent microtubules and estimate the real space error in the reproducibility experiment to be on the order of only a few percent (e.g. ~1 nm in the presented case, now Supplementary Figure 19).

In addition to these points, a comparison to state of the art techniques capable of imaging deep into tissue should be included in the Discussion section. This should include localisation techniques such as astigmatism, biplane, double-helix point-spread-function and multi-focal microscopy when used with selective plane illumination capable of imaging to a similar depth as presented in this work (HILO and light-sheet variations) as well as lattice light sheet. The discussion should include aspects such as signal to background level, depth of field, applicability to labelling methods, acquisition speed and ease of use.

We have extended our section in the discussion part of the manuscript.

Major recommendations to current manuscript:

1. It is unfair to compare TIRF and HILO illumination with the same statement of 'only offering limited sample penetration depths' in the first paragraph of the introduction. TIRF illumination is severely limited to depths of ~500nm whereas HILO is capable of facilitating 3D single-molecule imaging at depths >10 μ m

without the requirement for PAINT labelling methods. Please clarify this statement with appropriate data or reference.

We agree with the reviewer on this and have now made the distinction between TIRF and HILO more clear.

2. In the second paragraph of the introduction, the discussion of reduced excitation intensity and ability to switch dyes into the dark state is only applicable to STORM dyes and not PALM dyes. SDC illumination is not ideal for PALM imaging, but this is due to bleaching of molecules out of the focal plane, resulting in a large fraction of 'missed' localisations. This point should be clarified and extended to include limitations of the SDC system for imaging PALM dyes.

We thank you for pointing out this important issue. We have now added this point to the introduction.

3. In the first paragraph of the Discussion section, it is stated that 'imaging performance is maintained over the full thickness of a mammalian cell' although no metrics of performance have been evaluated as a function of axial depth into the sample. As mentioned in the above section, this analysis must be present in order for such claims to be made. Further quantification is needed to support this statement or it should be removed entirely.

We do apologize for the imprecise statement. We have now included quantification with regard to achievable localization precision at depths and have rephrased the manuscript accordingly.

4. In figure 1, errors should be quoted and explained with the measurements presented in d and i.

We now include different metrics for showing achievable localization precisions (e.g. NeNA vs. single site averaged standard deviation from sum images – itself a measure for error – in the 20 nm and tetrahedron dataset) and show that they are in good agreement with each other.

Minor recommendations to current manuscript:

1. 'TIRF' is a more accepted acronym compared to 'TIR' please change 'TIR' to 'TIRF' throughout the text.
2. At certain places throughout the text the vocabulary points to a biased position on the subject.

Examples of this include but are not limited to:

- a. The reference to PAINT techniques as 'easy-to-implement' in the introduction, as this is not necessarily the case in cell imaging.
- b. The reference to SDC as a 'promising alternative' to selective plane illumination in the introduction.
- c. The reference to SDC detection as 'Essentially wide-field' in the introduction.
- d. 'Encouraged by this result' in the Results section.
- e. 'in excellent agreement' in the Results section.
- f. 'Buoyed by our success' in the Results section.
- g. 'demonstrating exquisite optical sectioning' in the Results section.
- h. 'lattice light-sheet and 4Pi microscopy present excellent approaches' in the Discussion section.
- i. 'making it a truly 'all-in-one' instrument' in the Discussion section.

Please change any biased or leading terminology to be more impartial, with special emphasis on examples f, g and i.

We thank the reviewer for this suggestion and apologize for choosing potentially misleading terminology. We have now rephrased parts of the manuscript accordingly.

In conclusion we want to thank this reviewer again for raising important points that triggered more detailed analysis of SDC-PAINT, which helped us to improve the manuscript.

Reviewers' Comments:

Reviewer #1 is satisfied with your revision, as indicated in the Remarks to the Editors section.

Reviewer #2 (Remarks to the Author):

The authors have satisfactorily addressed my concerns.

I'm particularly pleased that the discussion of how the technique compares with other methods seems much more objective.

Reviewer #3 (Remarks to the Author):

Response:

Thank you for addressing all of the concerns presented in the original review so convincingly. I believe that these changes have resulted in a dramatic improvement to the manuscript, to the point that it is now suitable for publication in Nature Communications.

There are a few minor changes that I would like to be addressed:

1. In the second paragraph of the Introduction a sentence should read, "While this is less of a concern for PALM microscopy, bleaching of fluorescent proteins outside of the detection volume prior to detection in the confocal volume will here ultimately reduce the number of detectable proteins. "
2. In addition to figures S15-18, showing raw data, I would like to see two additional aspects of the raw data:
 - a. An SI movie representing a typical acquisition of 3D SDC-PAINT raw data (this need only be a few hundred frames).
 - b. A few representative PSFs being highlighted with zoomed-in cutouts in figures S15-18, as the large field of view makes examining individual PSFs difficult.
3. Please plot figure S4 as two 2D figures with each precision against detected photons (or as one 2D figure with both plots highlighted in different colours) as opposed to the 3D plot that is difficult to read.

These changes should not be too much work and once implemented the authors should be proud of this work.

Reviewer #3 (Remarks to the Author):

Thank you for addressing all of the concerns presented in the original review so convincingly. I believe that these changes have resulted in a dramatic improvement to the manuscript, to the point that it is now suitable for publication in Nature Communications.

There are a few minor changes that I would like to be addressed:

1. In the second paragraph of the Introduction a sentence should read, “While this is less of a concern for PALM microscopy, bleaching of fluorescent proteins outside of the detection volume prior to detection in the confocal volume will here ultimately reduce the number of detectable proteins. “
2. In addition to figures S15-18, showing raw data, I would like to see two additional aspects of the raw data:
 - a. An SI movie representing a typical acquisition of 3D SDC-PAINT raw data (this need only be a few hundred frames).
 - b. A few representative PSFs being highlighted with zoomed-in cutouts in figures S15-18, as the large field of view makes examining individual PSFs difficult.
3. Please plot figure S4 as two 2D figures with each precision against detected photons (or as one 2D figure with both plots highlighted in different colours) as opposed to the 3D plot that is difficult to read.

These changes should not be too much work and once implemented the authors should be proud of this work.

We thank the reviewer for the positive evaluation of our resubmitted manuscript. We have addressed the remaining points in the revised manuscript.